# Pairing mechanism in the ferromagnetic superconductor UCoGe

Beilun Wu[1], Gaël Bastien[1], Mathieu Taupin[1,2], Carley Paulsen[3], Ludovic Howald[1,4], Dai Aoki[1,5] & Jean-Pascal Brison[1]

Superconductivity is a unique manifestation of quantum mechanics on a macroscopic scale, and one of the rare examples of many-body phenomena that can be explained by predictive, quantitative theories. The superconducting ground state is described as a condensate of Cooper pairs, and a major challenge has been to understand which mechanisms could lead to a bound state between two electrons, despite the large Coulomb repulsion. An even bigger challenge is to identify experimentally this pairing mechanism, notably in unconventional superconductors dominated by strong electronic correlations, like in high-Tc cuprates, iron pnictides or heavy-fermion compounds. Here we show that in the ferromagnetic superconductor UCoGe, the field dependence of the pairing strength influences dramatically its macroscopic properties like the superconducting upper critical field, in a way that can be quantitatively understood. This provides a simple demonstration of the dominant role of ferromagnetic spin fluctuations in the pairing mechanism.

[1] Université Grenoble Alpes, CEA, INAC-PHELIQS, F-38000 Grenoble, France. [2] Institute of Solid State Physics, Vienna University of Technology, Wiedner Hauptstrasse 8-10, Vienna 1040, Austria. [3] Université Grenoble Alpes, CNRS, F-38000 Grenoble, France. [4] Swiss Light Source, Paul Scherrer Institut, CH-5232 Villigen PSI, Switzerland. [5] Institute for Materials Research, Tohoku University, Oarai, Ibaraki 311-1313, Japan. Correspondence and requests for materials should be addressed to J.-P.B. (email: jean-pascal.brison@cea.fr).

In the so-called conventional superconductors, the pairing interaction has been identified as arising from the coupling between electrons and the lattice, which can overcome the Coulomb repulsion because of retardation effects. The extensive knowledge of the electron–phonon interaction in conventional metals, has allowed a complete description of the pairing mechanism and its signatures on the superconducting properties, already back in the sixties[1]. However, even with these powerful theoretical developments, predicting from first principles a precise value of the superconducting critical temperature ($T_{sc}$) remains a formidable task. This is due to the exponential dependence of $T_{sc}$ on the strength of the pairing interaction, and to the difficulties in computing the screened-retarded Coulomb repulsion potential. As a result, the most convincing demonstration that the pairing arises from the electron–phonon coupling in conventional superconductors does not come from the predicted values of $T_{sc}$: it comes from measurements and analysis of phonon anomalies in the tunnelling spectra[2], which have been quantitatively explained by the Eliashberg theory[1].

In strongly correlated electron systems, where the pairing mechanism may arise from magnetic or other exotic degrees of freedom, the situation is much more complicated. No equivalence of the Eliashberg theory can describe accurately the normal state of these systems. In most cases, there is not even a clear separation between electronic quasiparticles (if they exist), and excitations mediating the pairing (usually over-damped, instead of being well defined modes as for phonons), as they both originate from the same electronic degrees of freedom.

Thus for these unconventional superconductors, the main method to get some insight on the pairing mechanism remains to force a change of its strength, and to monitor the corresponding change of $T_{sc}$. This is typically what is done in conventional superconductors with the isotope effect, where the change of $T_{sc}$ with ion masses can be related to the change of phonon frequencies. In strongly correlated electron systems, one can use pressure or doping to tune electronic instabilities and provoke the appearance of a superconducting phase. This was first realized with heavy fermion superconductors[3,4], revealing that superconductivity appears frequently at the verge of a magnetic instability. Similar behaviour has been identified in organic, cuprates and iron pnictides superconductors[5–8]. In heavy fermion systems, the problem has also been tackled by comparisons between the pairing strength and normal state quantities, when they are changed under pressure[9] or magnetic field[10].

In this study, we investigate the pairing strength of the ferromagnetic superconductor UCoGe. Macroscopic homogeneous coexistence of ferromagnetism and superconductivity has been clearly established[11–13] in three uranium based systems: UGe$_2$ (ref. 14) under pressure, URhGe[15] and UCoGe[16] at ambient pressure. The last two, URhGe and UCoGe, show similar properties in many ways. They have the same orthorhombic crystal structure (space group Pnma), and are both weak itinerant ferromagnets, with a strong uniaxial anisotropy along the easy magnetization axis (c-axis). However, their upper critical fields ($H_{c2}$) present some remarkable differences. We will show that these differences, and some anomalous properties of $H_{c2}$ in UCoGe, can be understood within a general theoretical framework describing superconductivity arising from ferromagnetic fluctuations[17].

## Results

### Bulk determination of $H_{c2}$ in UCoGe with thermal condutivity.
UCoGe has a Curie temperature ($T_{Curie}$) of about 2.5 K and a bulk superconducting transition at $T_{sc} \approx 0.5$ K (ref. 16). For clean, well

oriented samples, $H_{c2}$ along the b-axis shows an S-shape behaviour[18], which is reminiscent of the re-entrant superconducting phase observed in URhGe for the same field direction[19]: this suggests that in both cases, superconductivity is enhanced under magnetic field along the b-axis. However, there are other features of $H_{c2}$ which are unique in UCoGe, and very puzzling. At first, according to previous resistivity measurements, $H_{c2}$ is strongly anisotropic: it is > 20 times higher along the two transverse directions (a,b) than along the easy magnetization axis[18]. In addition to this strong anisotropy, the angular dependence of $H_{c2}(0)$ in the (a,c) plane is extremely sharp near the a-axis[20,21], at odds with the usual elliptical behaviour. Strong anisotropy in $H_{c2}$ is commonly observed in low dimensional superconductors, like organics[22]. or high-Tc cuprates[23,24]. However, the transport properties[21,25] and the almost spherical Fermi surface pocket observed in Shubnikov-de Haas measurements[25] suggest that the electronic structure of UCoGe is essentially isotropic in the normal phase.

In addition, the temperature dependence of $H_{c2}$ in UCoGe is marked by unusual positive curvatures along the three crystallographic directions[18,26]. These features call on confirmation by a bulk sensitive probe: the resistive superconducting transition under field occurs when vortex pinning is strong enough for the measurement current. So it has been found that in some organic superconductors[27] and in most cuprates[28,29], the resistive transition under field does not determine $H_{c2}$. Instead, it measures an irreversibility line, that might separate a vortex liquid and a vortex solid phase. This line can be much lower than $H_{c2}$, and it displays a strong upward curvature. Owing to the similarities of $H_{c2}$ in UCoGe and in these systems, we have used thermal conductivity as a probe (as in refs 27,29), to obtain the bulk $H_{c2}$ along the three crystallographic directions in UCoGe (see Fig. 1a,b for a zoom on the c-axis).

Figure 1b shows that both bulk and resistive measurements along the c-axis display the same upward (positive) curvature, with the absence of saturation down to the lowest temperatures (10 mK). At very low field, the bulk $H_{c2}$ is lower than the resistive determination, but the agreement is much better at higher field, particularly for the criterium of zero resistance. This complies with the usual sensitivity of the resistive transition to filamentary superconducting paths, rapidly suppressed under field. Altogether, the anisotropy and the anomalous temperature dependence of $H_{c2}$ in UCoGe reported in previous resistivity studies[18], are well confirmed by our bulk measurements, which excludes an explanation of these anomalies by an irreversibility line.

A specific feature of UCoGe is the possible role of ferromagnetic fluctuations in the pairing mechanism, and their rapid suppression by magnetic fields applied along the c-axis. This suppression has been previously demonstrated by NMR studies, which revealed strong Ising-type longitudinal magnetic fluctuations in UCoGe, with a very anisotropic response to an applied magnetic field[21]. It has also been confirmed by electrical and thermal transport[30]. A first empirical model for the effects of a field dependence of the superconducting pairing strength has been derived from the NMR response[21,31]. It could qualitatively account for the $H_{c2}$ anisotropy between the c and a-axis, but it predicts a vanishing slope of $H_{c2}$ at $T_{sc}$ along the c-axis, which is not observed (Fig. 1b). References 17,32 proposed an earlier theoretical approach based on a general and well defined framework. It had predicted that when pairing is mediated by ferromagnetic spin fluctuations, the field dependence of the magnetization or of the Curie temperature drives the field dependence of the superconducting coupling strength. In the following sections we discuss this idea of a field-dependent pairing mechanism, probed by the field dependence of normal-

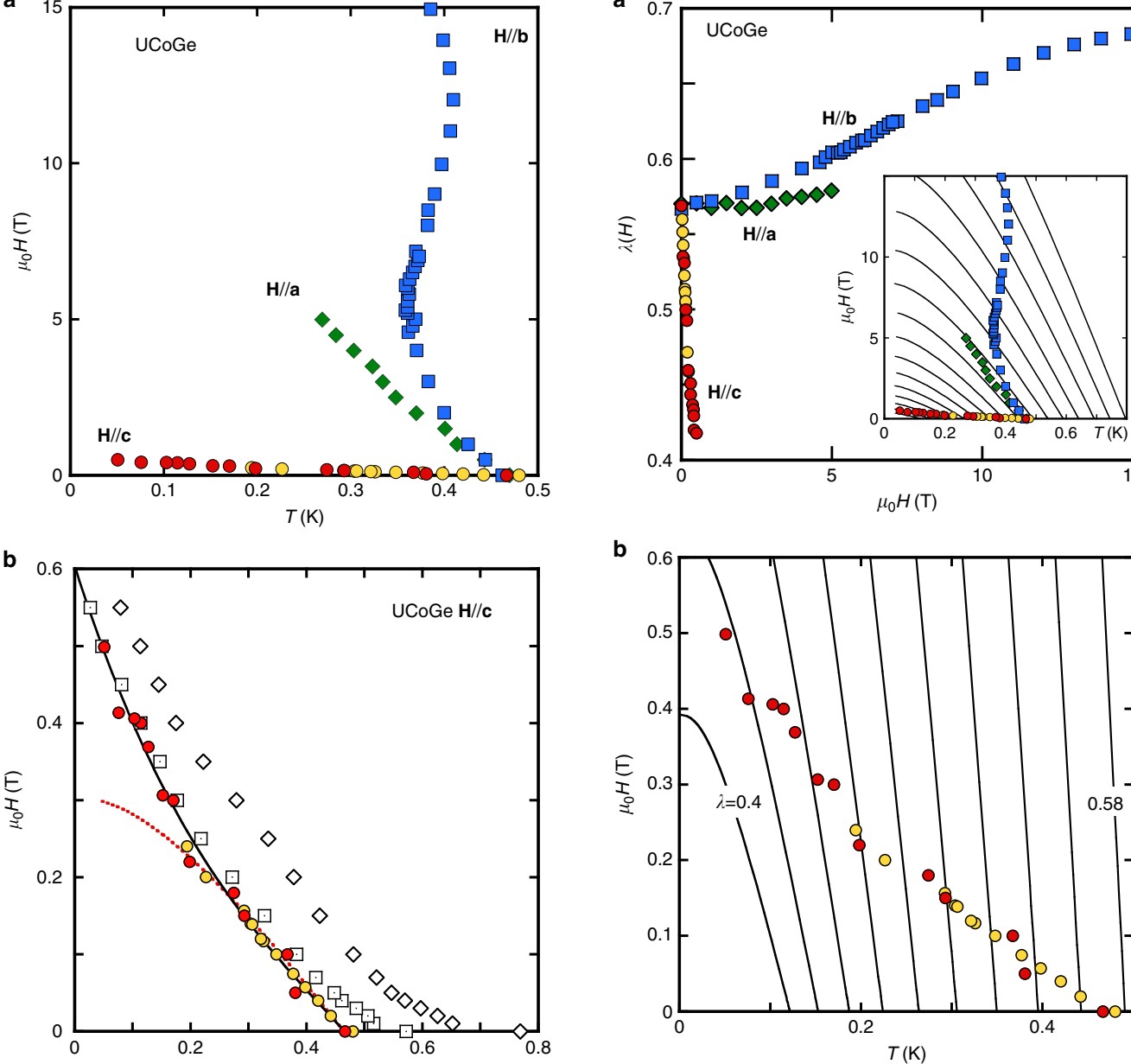

**Figure 1 | $H_{c2}$ in UCoGe from a bulk sensitive probe.** (**a**) $H_{c2}$ for the three principal axis of UCoGe, from thermal conductivity measurements. The S-shape of $H_{c2}$ for **H//b**-axis is well visible, as well as the enormous anisotropy between the **c**-axis, and the (**a**,**b**) plane. (**b**) Zoom along the **c**-axis. Red circles: from thermal conductivity; Yellow circles: from ac-susceptibility; Diamonds: from the onset of the resistive transition; Squares: from the resistive transition at $\rho = 0$; Solid line: guide to the eyes, showing the strong upward curvature of $H_{c2}$. Dotted line: by contrast, the classical $H_{c2}$ behaviour with negative curvature.

**Figure 2 | Field dependence of the pairing strength extracted from $H_{c2}$.** (**a**) $\lambda(H)$ for the three crystallographic directions. The inset shows the calculated $H_{c2}$ curves using the strong coupling model, each with a fixed coupling constant $\lambda$ from 0.4 up to 0.7 in steps of 0.02 from left to right, together with the data of Fig. 1a. (**b**) zoom of the inset of **a** for **H//c**. The $H_{c2}$ lines are almost vertical, visualizing the negligible role of the orbital limitation in the determination of $H_{c2}$ for this field direction.

state properties, with emphasis on the field direction **H//c**. To perform quantitative comparison to theoretical predictions[17], we measured specific heat, magnetization and ac susceptibility for this field direction, on the same single crystal. The raw data are presented in Methods Fig. 7a and in Supplementary Figs 2 and 3.

**Analysis.** To get an idea about the effect of a field dependence of the pairing strength on $H_{c2}$, a simple back-of-the-envelop calculation can be useful. For a superconductor in a weak coupling scheme, $T_{sc}$ is given by:

$$T_{sc} \sim \Omega \exp\left(-\frac{1}{\lambda - \mu^*}\right) \qquad (1)$$

$\Omega$ is the characteristic frequency of the pairing interaction (proportional to the Debye temperature for electron-phonon coupling), $\lambda$ is the pairing strength, which can now vary under field ($\lambda = \lambda(H)$), and $\mu^*$ is the Coulomb repulsion parameter (usually of order 0.1–0.15). Like all heavy fermion

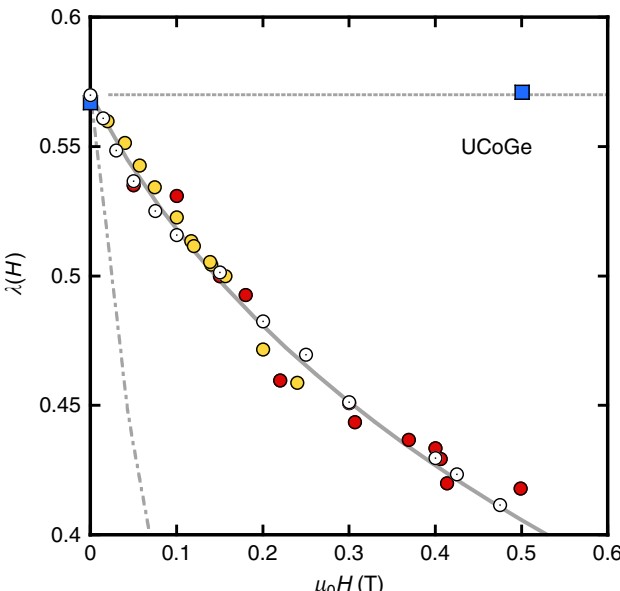

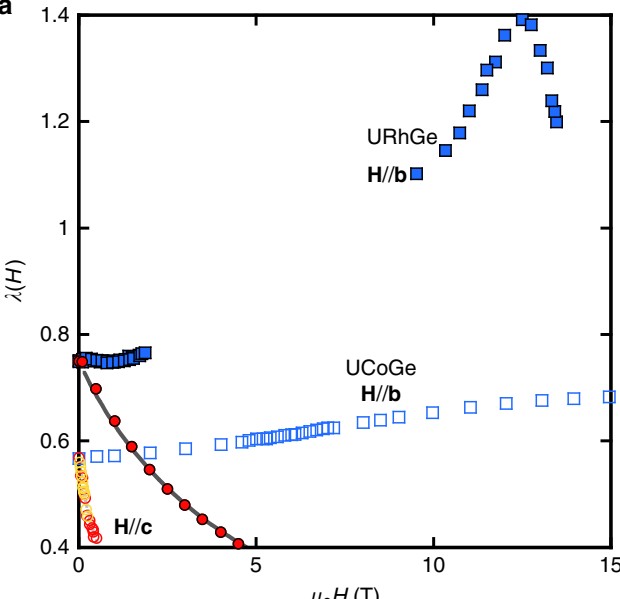

**Figure 3 | Analysis of the field dependence of the pairing strength for H//c in UCoGe.** Filled circles: from the experimental $H_{c2}$ curve. Open circles: from specific heat measurements. Lines: from equation (6), based on magnetization measurements performed on the same sample: solid line, for the optimized value of $\xi_{mag}k_F = 3.2$; dash-dotted line for $\xi_{mag}k_F = 1$ (localized magnetism). The low field regime of $\lambda(H)$ for **H//b** is also presented. Squares: $\lambda$ from $H_{c2}$. Doted line: prediction from equation (6) based on the measured field variation of $T_{Curie}$.

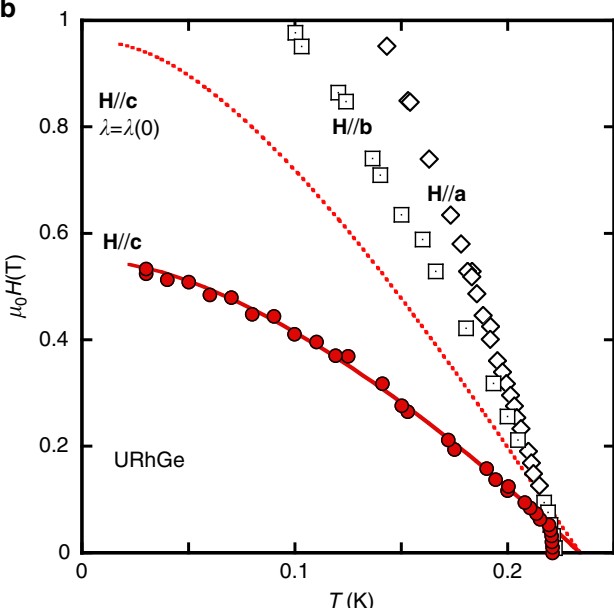

**Figure 4 | Analysis of $H_{c2}$ of URhGe.** (**a**) closed symbols, $\lambda(H)$ in URhGe along **b,c**-axis. Data along **b** extracted from $H_{c2}$, data along **c** extracted from the specific heat of ref. 44 (see Methods). Solid line: $\lambda(H)$ predicted by theory in ref. 17 with $\xi_{mag}k_F \sim 1$. Open symbols: $\lambda(H)$ in UCoGe (same as in Fig. 2a) for comparison. (**b**) Solid line: $H_{c2}//$**c** in URhGe based on $\lambda(H)$ obtained from the specific heat (see Fig. 4a). Broken line: for comparison, $H_{c2}$ with $\lambda$ fixed at $\lambda(0)$ (40% larger). Experimental data are from ref. 35. Circles: **H//c**; Squares: **H//b**; Diamonds: **H//a**. Note the much weaker anisotropy of $H_{c2}$ in URhGe compared with UCoGe (displayed on Fig. 1a).

superconductors, UCoGe is in the clean limit. According to standard Ginzburg-Landau theory, for a clean single-band superconducting system dominated by the orbital limitation, the upper critical field near $T_{sc}$ is given by:

$$H_{c2}^{orb} \sim \frac{\Phi_0}{2\pi\xi^2(T)} \sim \frac{\Phi_0}{3\xi_0^2}\left(1 - \frac{T}{T_{sc}}\right)$$
$$\sim \frac{\Phi_0 k_B^2}{0.1(\hbar\langle\nu_F\rangle)^2} T_{sc}(T_{sc} - T) \qquad (2)$$

where we used the Bardeen–Cooper–Schrieffer (BCS) expressions for the superconducting coherence length ($\xi(T) = 0.7\xi_0$ $(1 - T/T_{sc})^{-1/2}$, $\xi_0 = 0.18\frac{\hbar\langle\nu_F\rangle}{k_B T_{sc}}$, and $\langle\nu_F\rangle$ is the average Fermi velocity perpendicular to the field).

Taking account of the additional field dependence of $\lambda(H)$, the initial slope of $H_{c2}$ $(dH_{c2}/dT|_{T=T_{sc}})$ can be calculated from equation (2):

$$\frac{dH_{c2}}{dT}\bigg|_{T=T_{sc}} = \left(\frac{dT}{dH}\bigg|_{orb} + \frac{dT_{sc}}{d\lambda}\frac{d\lambda}{dH}\right)^{-1} \qquad (3)$$

According to equation (3), $dH_{c2}/dT|_{T=T_{sc}}$ is determined both by the usual orbital limit (first term on the right side), and by an additional term $\left(\frac{dT_{sc}}{d\lambda}\frac{d\lambda}{dH}\right)$ arising from the field dependence of $\lambda$. The idea for UCoGe is that this second term could be dominant: a large negative $\frac{d\lambda}{dH}$ would lead to a much reduced initial slope, so that the anisotropy of $H_{c2}$ could just reflect that of the field suppression of the ferromagnetic fluctuations. Another consequence of this dominant term is that the orbital effect would play little role in the temperature dependence of $H_{c2}$, opening new routes to explain the data of Fig. 1.

To study quantitatively this hypothesis of a field-dependent pairing strength, we choose an new angle of attack: instead of searching for a model that can reproduce $H_{c2}(T)$, we extract from the experimental data of Fig. 1a the field and direction dependence of $\lambda$ required to reproduce them. For

this, we calculate $H_{c2}$ for a series of fixed values of the pairing strength $\lambda$, with the help of a simple strong-coupling model for the upper critical field[33] described in ref. 9 (details in the Methods). The parameters of this model are the same as those of equations (1 and 2), but $T_{sc}$ and $H_{c2}$ are now calculated from a microscopic model, which also includes the effective mass $m^\star$ (or equivalently, the Fermi velocity) renormalization by the pairing interactions derived from

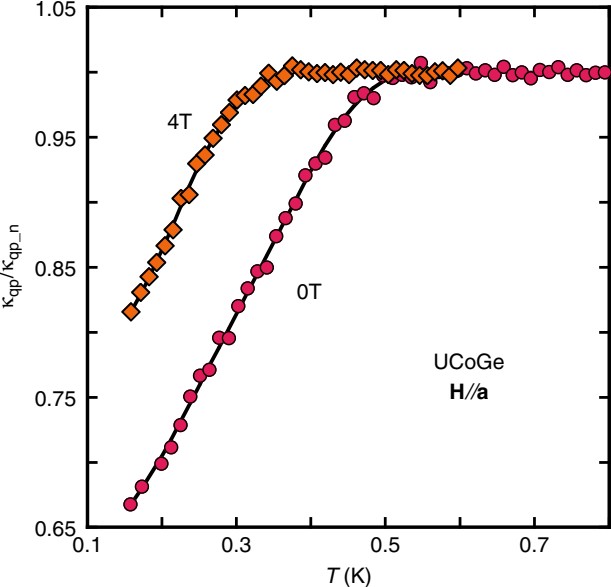

**Figure 5 | Superconducting transition in thermal transport.** Quasiparticle contributions to the thermal conductivity, normalized to its normal phase value, $\kappa_{qp}/\kappa_{qp\_n}$, calculated from thermal conductivity and normal state resistivity data, by assuming the validity of the Wiedemann–Franz law, and by eliminating the other contributions to the thermal transport. Magenta circles: $\kappa_{qp}/\kappa_{qp\_n}$ data for $\mu_0 H = 0$ T; orange diamonds: $\mu_0 H = 4$ T. The solid lines show the corresponding fits.

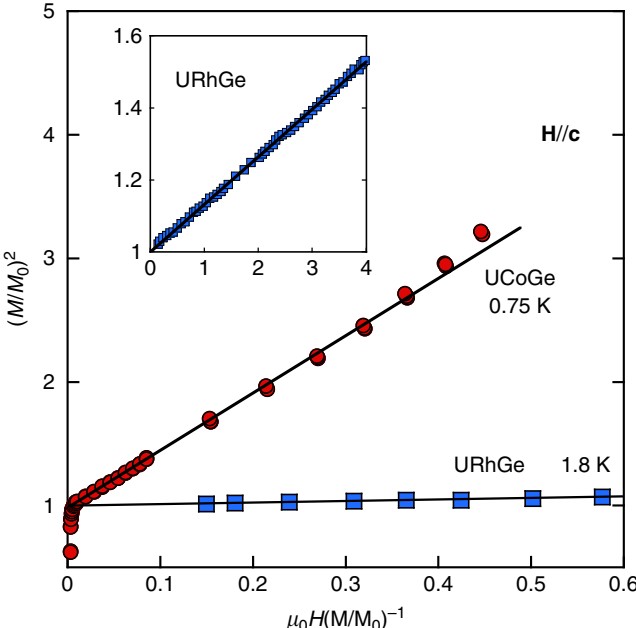

**Figure 6 | Comparison of Arrott plot between UCoGe and URhGe.** Arrott plot for UCoGe for field along **c**-axis, at $T = 0.75$ K (red circles). Deviation from the linear behaviour starts at field above 0.6 T. Arrott plot for URhGe in the same field direction is also presented for comparison (blue triangles), at $T = 1.8$ K. The inset shows the curve of URhGe in the complete field range[44] (same labels and units as the main plot). The solid straight lines are simple guides to the eyes.

Eliashberg equations[1,33]:

$$m^* = m_{\text{band}}(1 + \lambda)$$
$$v_{\text{F}} = \frac{v_{\text{F}}^{\text{band}}}{1 + \lambda} \qquad (4)$$

where $m_{\text{band}}$ $\left(v_{\text{F}}^{\text{band}}\right)$ is the band mass of the quasiparticles (their band Fermi velocity), renormalized by all the interactions apart from the pairing interactions. Then we compare our experimental data to the series of calculated $H_{c2}$ (inset in Fig. 2a) and extract $\lambda(H)$ needed to reproduce $H_{c2}$ in the three directions (Fig. 2a). The two major assumptions used in this process are that UCoGe in the normal phase is isotropic, and that the influence of $p$-wave pairing on $H_{c2}$ is approximated by a calculation for $s$-wave superconductors, with no paramagnetic limitation. For the first point, it assumes equal average Fermi velocities $\langle v_{\text{F}}^i \rangle$ along each axis $i$, so that the $H_{c2}$ anisotropy arises only from the difference in $\lambda(H)$ between **a**, **b** and **c**-axis. For the second point, some anisotropy and small differences in the temperature dependence of $H_{c2}$ could also come from the exact form of the $p$-wave order parameter[34], as discussed in URhGe[35]. These effects are neglected here, as in UCoGe, anisotropy originating from superconducting gap nodes has not been detected so far, even on the best available samples[36].

As seen in Fig. 2a, for fields along the **a**-axis, $\lambda$ is essentially constant up to 5 T with only a slight increase of order 1.5% induced by the visible upward curvature in Fig. 1a. For **H//b**, the pronounced S-shape of $H_{c2}$ gives rise to a monotonous $\sim 20\%$ increase of $\lambda$ up to 15 T. The most striking feature is the sharp decrease of $\lambda$ for **H//c**, of nearly 30%, in a narrow field range between 0 and 0.5 T. Figure 2b zooms on the data for **H//c**, and the series of calculated $H_{c2}$ lines: they are almost vertical, meaning that $H_{c2}$ is almost entirely controlled by $T_{sc}(\lambda(H))$, not by the orbital limitation. This justifies *a posteriori* the assumption of isotropic $\langle v_{\text{F}}^i \rangle$ in zero field.

The determination of $\lambda(H)$ from $H_{c2}$ depends quantitatively on the choice of the zero field value $\lambda(0)$. It has been fixed by comparison with the variation of the Sommerfeld coefficient $\gamma$ of the specific heat ($C_p$, $\gamma = C_p/T$) for **H//c**. Indeed, the renormalization of the effective mass (equation (4)) leads to a renormalization of $\gamma \propto m^* \propto (1 + \lambda)$. So the field variation of $\lambda$ reported in Fig. 2a should be reflected in a field variation of $\gamma$ as:

$$\lambda(H) = \frac{\gamma(H)}{\gamma(0)}(1 + \lambda(0)) - 1 \qquad (5)$$

Figure 3 shows that, for the value $\lambda(0) = 0.57$, a satisfactory agreement is reached between $\lambda(H)$ obtained from the strong-coupling analysis of $H_{c2}$, and $\lambda(H)$ deduced from our data of $\gamma(H)$ according to equation (5) (see Methods Fig. 7b). In both cases, there is an initial linear-decrease of $\lambda$ at low field, as opposed to the $\sqrt{H}$ dependence of $d\lambda/dH$ proposed in the pioneering work ref. 21. Experimentally, because of the torque exerted by the spontaneous magnetization under perpendicular field orientation, there are still no measurements of $\gamma$ under fields along **b**- or **a**-axis. However, an increase of $\lambda$ for **H//b** is qualitatively expected from the coefficient of the inelastic contribution to the resistivity ($AT^2$ term), which shows a maximum near 14 T (ref. 18).

**Theoretical framework.** Several theoretical studies have been proposed to describe the emergence of superconductivity in a ferromagnetic background[17,37–42]. All these models suggest a pairing mechanism based on spin-spin interactions proportional to the dynamical spin susceptibility. In ref. 17 (extended in the more recent ref. 32), a field dependence of the pairing strength is predicted from a general framework of the ferromagnetic state (Landau approach), valid both in the

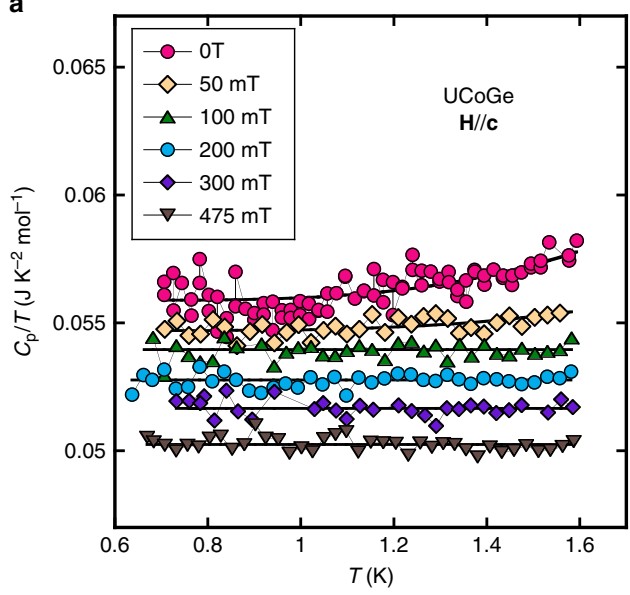

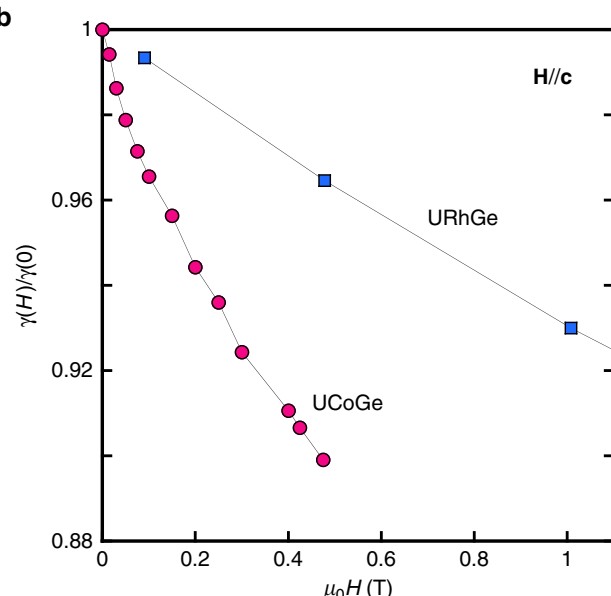

**Figure 7 | Specific heat in UCoGe and URhGe.** (**a**) Temperature dependence of $C_p/T$ of UCoGe, for **H//c**. For magnetic fields close to 0 T, a small increase in $C_p/T$ occurs when $T$ approaches $T_{Curie}$. The $C_p/T$ curves are then fitted with a function $C_p/T = \gamma + \delta$. $\exp(-T_0/T)$ to obtain the associated Sommerfeld coefficient $\gamma$. (**b**) Field dependence of the Sommerfeld coefficient $\gamma$ in UCoGe (magenta circles) for **H//c**, normalized to its zero field value. The same data for URhGe[44] on the same field range, equally for **H//c**, are presented for comparison, with the blue squares.

itinerant and localized limits. It has been derived in the weak coupling limit for superconductivity: comparing this theory to our experimental results implicitly assumes that the same field dependence is preserved in the intermediate coupling regime, and shows up in normal state quantities like $\gamma(H)$. Another assumption is that the expression of the Landau free energy, valid at temperatures close to $T_{Curie}$, is still valid near $T_{sc}$. This hypothesis is supported by our magnetization (M) data: at 0.5 K, they are straight lines on an Arrott plot (see Methods Fig. 6).

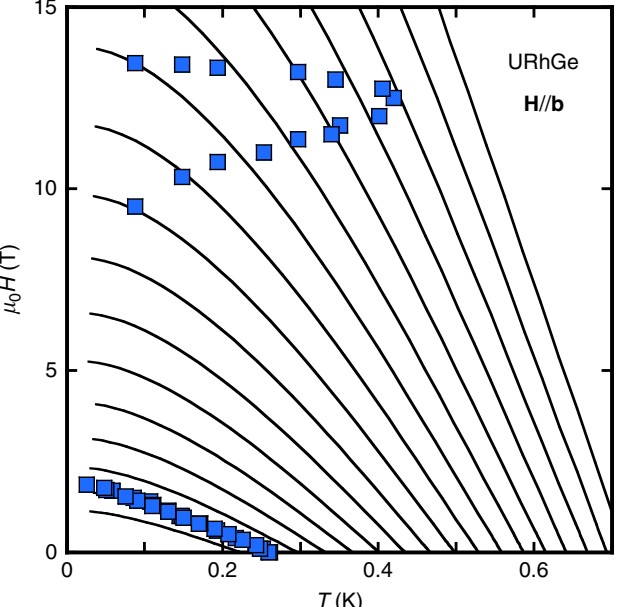

**Figure 8 | Strong coupling calculation of $H_{c2}$ in URhGe.** Blue squares: data of $H_{c2}$//**b** in URhGe from ref. 45. Full lines: series of calculations of $H_{c2}$ with fixed values of the strong coupling constant $\lambda$, which changes from 0.7 to 1.5 by steps of 0.05. $\Omega$ and $v_F^{band}$ have been adjusted so as to match the experimental $T_{sc}$ and initial slope for $\lambda = 0.75$ and $\mu^\star \sim 0.1$. Note that if $\lambda$ had, in zero field, the value it reaches at $H_R$ ($\lambda(H_R) \sim 1.4$), $T_{sc}$ would be of order 0.67 K instead of 0.26 K.

Reference 32 gives expressions for the susceptibilities $\chi_{ij}(T, \mathbf{M}, \mathbf{q})$, with a wave vector ($\mathbf{q}$) dependence arising from the exchange terms in the Landau free energy. The pairing strength for a $p$-wave order parameter (called 'g' instead of '$\lambda$' in ref. 32) is calculated in a form which can be cast as:

$$\lambda(H) = \lambda(0) \frac{(1+a^2)^2}{(\Theta+a^2)^2} \tag{6}$$

with $a = \xi_{mag} k_F$ a numerical parameter ($\xi_{mag}$ the magnetic coherence length associated to the ferromagnetic order and $k_F$ the Fermi wave vector), and the factor $\Theta$ given by

$$\begin{aligned} \Theta(\mathbf{H}//\mathbf{c}) &= \frac{1}{2}\left(3\frac{M_z^2}{M_0^2} - 1\right) \\ \Theta(\mathbf{H}\perp\mathbf{c}) &= \frac{T_{Curie}(H) - T_{sc}}{T_{Curie}(0) - T_{sc}} \end{aligned} \tag{7}$$

$M_z$ ($M_0$) is the (spontaneous) magnetization along the **c**-axis. This simple form is valid in a one-band approximation[17]: two-band effects arise automatically for $p$-wave - equal spin pairing states[43], but in practice they change very little the results, notably for **H//c** (see Supplementary Notes 1 and 2).

Equations (6 and 7) show that this general framework for ferromagnetic superconductors predicts a decrease of the pairing strength with field along the easy axis, controlled by the field dependence of the magnetization. Our magnetization data yield an excellent agreement between equation (6) and $\lambda(H)$ deduced from $H_{c2}$ and $\gamma(H)$ for a value of $\xi_{mag} k_F \sim 3.2$ (Fig. 3). The large value of $(\xi_{mag} k_F)^2 \sim 10$ damps the strong increase of $M/M_0$ (+60% from 0 to 0.5 T): otherwise, the predicted theoretical suppression of $\lambda$ would be much larger (see dash-dotted line for $\xi_{mag} k_F \sim 1$ in Fig. 3). This large value of $\xi_{mag}$ points to the itinerant nature of the ferromagnetism in this compound: for a localized magnetic system, one would expect $\xi_{mag}$ of the order of interatomic distances, so $\xi_{mag} k_F \sim 1$.

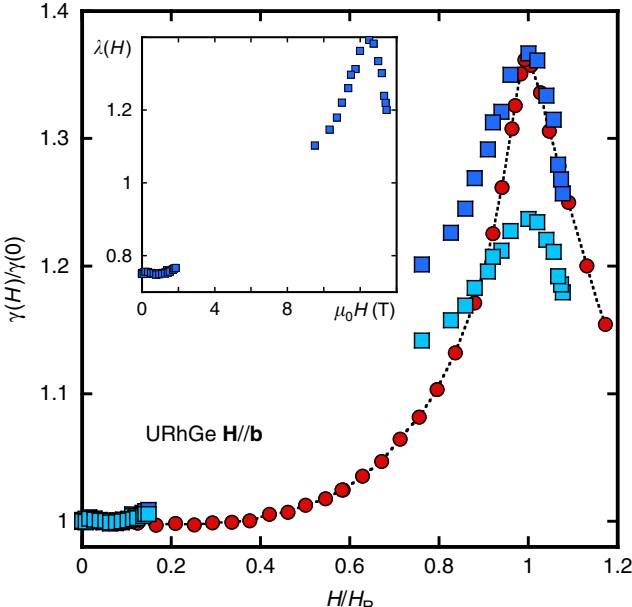

**Figure 9 | Adjustment of $\lambda_0$ in URhGe.** Red circles: specific heat Sommerfeld coefficient for **H**//**b** from ref. 44, normalized to its value at zero field. Dark blue squares: ratio $(1 + \lambda(H))/(1 + \lambda(0))$ deduced from the calculations of $H_{c2}$ presented in Fig. 8 for $\lambda(0) = 0.75$. Light blue squares: same ratio, but with the calculations of $H_{c2}$ for $\lambda(0) = 0.6$. It shows that the amplitude of variation of the specific heat coefficient is a very selective criterium for the determination of $\lambda(0)$ in this system. Inset: Corresponding variation of $\lambda$, for $\lambda(0) = 0.75$.

For **H** along **a** and **b**-axis, the field dependence of $\lambda$ in equations (6 and 7) is related to the change of $T_{\text{Curie}}$, which has been determined experimentally in ref. 18. In the same field range (0–1 T), there is essentially no detectable change of $T_{\text{Curie}}$, and so no change of $\lambda$ for **H**//**b** or **H**//**a**, in agreement with the results in Fig. 2a. At higher fields (along **b**-axis), the situation becomes much more complicated (see Supplementary Note 3 for a detailed discussion). Quantitative agreement between theory and experiment is hard to reach in this case, suggesting an inadequacy of the Landau framework to describe the evolution of the magnetic state in the transverse field configuration. For example, the emergence of a soft magnon mode has been proposed in another work to account for the re-entrant phase in URhGe for **H**//**b** (ref. 42), an effect that could not be included in the present theory.

## Discussion

Apart from the high-field behaviour along the transverse directions, the theoretical framework in ref. 32 successfully explains the positive curvature of $H_{c2}$ in UCoGe for **H**//**c**, as well as the anomalous anisotropy between **c**-axis and the (**a**,**b**) plane. This theory includes no element specific to UCoGe, and should apply equally to URhGe. The validity of our claims for UCoGe is challenged by the usual negative curvatures and the small anisotropy of $H_{c2}$ in URhGe[35].

A major difference between the two systems is that in URhGe, $M_z/M_0$ has a much weaker field dependence for **H**//**c** than in UCoGe[44] ($\sim$3% instead of 60% in UCoGe, at 0.5 T). This is the key element to understand, according to equation (6), the smaller suppression of $\lambda$ for **H**//**c**. Quantitatively, we can take advantage of the specific heat data available for **H**//**b** in this system[44], to determine the value of $\lambda(0) \sim 0.75$ from comparison to $H_{c2}$ (ref. 45) (see Methods Fig. 9). Once $\lambda(0)$ is fixed, the specific heat

data for **H** along the **c**-axis[44] determines how $\lambda$ changes with field according to equation (5). It compares successfully to the theoretical predictions of equation (6), and the magnetization data[44] along the **c**-axis, in a wide field range, for $\xi_{\text{mag}} k_F \sim 1$ (see Fig. 4a). This value of $\xi_{\text{mag}} \sim k_F^{-1}$ indicates that magnetism in URhGe lies close to the localized limit, in accordance with the larger ordered moment and the net spin rotation induced by fields along the **b**-axis[19]. Figure 4a compares $\lambda(H)$ along **c** and **b**-axis for URhGe and UCoGe: clearly, apart from the sharp maximum in URhGe at 12 T corresponding to the moment rotation[19], the difference between the two systems is just quantitative.

Figure 4b displays $H_{c2}$ for **H**//**c** in URhGe calculated with $\lambda(H)$ deduced from $\gamma(H)$, together with the data of ref. 35 (the Fermi velocity is adjusted to match the slope of $H_{c2}$ at $T_{sc}$, and $T_{sc}$ has been shifted to take account of the internal field arising from $M_0$ (ref. 35)). Compared with the $H_{c2}$ calculated with $\lambda$ fixed at its zero-field value (dotted line in Fig. 4b), the suppression of $\lambda(H)$ for **H**//**c** leads to a 40% reduction of $H_{c2}(0)$: it is simply not strong enough to inverse the curvature of $H_{c2}$.

Altogether, these results prove that in UCoGe, and most likely in URhGe, the pairing mechanism does arise from ferromagnetic spin-fluctuations. The demonstration has been possible because of the particularly strong variation of the pairing strength with magnetic field in UCoGe, controlling the anomalous temperature dependence of $H_{c2}$ (along the easy axis) and its enormous anisotropy. These robust experimental features of the superconducting phase are quantitatively related to the field variation of normal state properties, and they are explained naturally with a general theoretical framework, without a need for a precise microscopic description of the complex ferromagnetic background. This leads to a solid and rare identification of an unconventional pairing mechanism among strongly correlated electron systems.

## Methods

**Sample.** UCoGe crystallizes in an orthorhombic structure, and the **c**-axis is the easy magnetization axis, the **a** and **b**-axis are the hard and intermediate axis, respectively. High-quality single crystals were grown by the Czochralski method in a tetra-arc furnace and further annealed: we have chosen a sample with a modest residual resistivity ratio (RRR $\approx$ 16), in order to have a good determination of $T_{sc}$ by thermal transport. Indeed for large RRR samples, the suppression of inelastic scattering at low temperatures leads to a large increase of the thermal transport ($\kappa$)[30], which masks the onset of the superconducting transition. By contrast, when the RRR is not as good, inelastic process do not dominate over elastic scattering, deviations from the Wiedemann-Franz law at $T_{sc}$ are smaller, and the occurence of the superconducting transition is marked by a clear kink on $\kappa(T)$.

**Measurement of $H_{c2}$ by thermal transport.** Thermal conductivity measurements were performed with the standard one-heater-two-thermometers method. The resistive heater and two thermometers were connected to the sample with 15 μm gold wires, spot-welded on the sample and glued with silver paste on the thermometer side. The temperature rise along the sample was set to be around 3% for all the measurements. The same gold-wire contacts were used for the four-wire ac-resistivity measurements, which permits to compare measurements from the two probes with exactly the same geometrical factor. For magnetic fields along **c**-axis, both thermal conductivity and resistivity were measured in a dilution refrigerator which can be cooled down to 10 mK. For magnetic fields along **a** and **b** axes, the measurements were performed in another dilution refrigerator down to 150 mK, in magnetic fields up to 15 T. The fact that $H_{c2}$ in UCoGe is extremely sensitive to any field component along the $c$ axis, makes precise field orientation a crucial issue: we are equipped with two piezo-goniometers (with perpendicular rotating axes) and one piezo-rotator (from Attocube). This allowed us to orient the field direction with a deviation less than 0.05° *in situ*, by following the angular dependence of resistivity in the width of the superconducting transition.

Supplementary Fig. 1 shows some of the thermal conductivity ($\kappa$) measurements in UCoGe for magnetic fields along the **a**-axis. For improved control and precision on the determination of the superconducting transition, we used a fitting procedure of the data, based on the following considerations:

— the thermal conductivity of metals is the sum of contributions from electrons (quasiparticles), phonons and in some cases, other bosonic contributions like magnons or more generally, magnetic fluctuations:

$$\kappa = \kappa_{\text{qp}} + \kappa_{\text{phonons}} + \kappa_{\text{magnons}} + \ldots \qquad (8)$$

For our sample with RRR $\sim 16$, we can apply the Wiedemann-Franz law to estimate the electronic contribution to the thermal conductivity from the resistivity data ($\rho$) in the normal phase. With $L_0 = 2.44.10^{-8}\,\mathrm{W\,\Omega\,K^{-2}}$ (the Lorentz number):

$$\frac{\kappa_{\mathrm{qp\_n}} \cdot \rho}{T} = L_0 \qquad (9)$$

— The ratio of the total thermal conductivity to the normal-phase electronic contribution to the thermal conductivity, is calculated as:

$$\frac{\kappa}{\kappa_{\mathrm{qp\_n}}} = \frac{\kappa \rho_n}{L_0 T} = \frac{\kappa_{\mathrm{qp}}}{\kappa_{\mathrm{qp\_n}}} + \frac{\kappa_{\mathrm{other}}}{\kappa_{\mathrm{qp\_n}}} \qquad (10)$$

The first term of equation (10) is the normalized electronic contribution, $\kappa_{\mathrm{qp}}/\kappa_{\mathrm{qp\_n}}$, which should equal 1 in the normal phase and decrease with temperature in the superconducting phase. The second term term arises from the other contributions (phonons, magnons, and so on), $\kappa_{\mathrm{other}}/\kappa_{\mathrm{qp\_n}}$, which should in principle exhibit little change in the neighbourhood of the superconducting transition.

— The temperature dependence of the ratio in equation (10) can be fitted as the sum of two functions: for the other contributions, we use a polynomial of order 3 that extrapolates to zero at $T = 0\,\mathrm{K}$: $F_{\mathrm{other}}(T) = aT + bT^2 + cT^3$; for the electronic part, we use a piecewise function $F_{\mathrm{qp}}(T)$, which equals 1 for $T > T_{\mathrm{sc}}$, and decreases linearly with $T$ below $T_{\mathrm{sc}}$. The superconducting transition temperature $T_{\mathrm{sc}}$ is a non-linear parameter of the fit. In practice, in order to improve the fit quality, we have introduced a (quadratic in temperature) smearing of the transition around $T_{\mathrm{sc}}$ and similarly for $T \rightarrow 0$. Figure 5 displays some of the $\kappa_{\mathrm{qp}}/\kappa_{\mathrm{qp\_n}}$ curves and the corresponding fits $F_{\mathrm{qp}}(T)$.

For the measurements for **H//b**, the fit works up to the highest measured field (15 T). However, for **H//a**, this determination of $T_{\mathrm{sc}}$ fails for fields above 5 T, essentially because the effect of the superconducting transition becomes weaker at these fields. For $H_{\mathrm{c2}}$ along the **c**-axis, the lowest temperature points were obtained with field sweeps: in such a case, the data were fitted directly with a piecewise function which takes linear temperature dependence both above and below the superconducting transition.

### Measurement of magnetization and ac-susceptibility.

Measurements of the magnetization (Supplementary Fig. 2) and ac-susceptibility (Supplementary Fig. 3) were made along **c**-axis using the high field, low temperature SQUID magnetometer developed at the Institut Néel in Grenoble. This magnetometer is equipped with a miniature dilution refrigerator capable of cooling the sample to below 100 mK. An 8 T superconducting coil supplies the dc field, and a small single layer copper coil is used for the ac field. A unique feature of the setup is that absolute values of the magnetization or susceptibility can be obtained by using the extraction method, without heating the sample.

For the easy magnetization axis, the phenomenological Landau theory used in ref. 32 predicts a linear behaviour in the Arrott plot: $2\alpha_z + 4\beta_z M_z^2 = H/M_z$, where $\alpha_z$ and $\beta_z$ are Landau coefficients. Figure 6 shows that such a behaviour is indeed followed experimentally in UCoGe, even at temperature close to the superconducting region, and for magnetic fields below 0.6 T, and above 3 mT, field above which a single magnetic domain is formed in the sample. Figure 6 also shows the same Arrott plot for URhGe (data from ref. 44), for **H//c**. It underlines that in UCoGe, the relative magnetization $M(H)/M_0$ changes much more strongly than in URhGe for fields along the easy axis. This is the main reason for the differences in the behaviour of $H_{\mathrm{c2}}$ along **c**-axis in the two systems.

### Measurement of specific heat.

The specific heat measurements were performed with a Physical Property Measurement System (PPMS-from Quantum Design), in a $^3$He cooling system, with the relaxation method. A temperature rise of about 3% was applied for each measurement, and the total measuring time was set to be five to six times the relaxation time. The raw results from the PPMS specific heat measurement program showed some anomalie because of the fits of the thermometer calibration. These anomalies could be suppressed by using a better fitting procedure for the thermometers' calibration (and for the Addenda raw data), so we worked on the PPMS log file, and re-performed the data analysis with the improved calibration laws.

Although the normal phase of UCoGe follows roughly the classical behaviours of a Fermi liquid (that is, for resistivity, $\rho = \rho_0 + AT^2$, and for specific heat, $C_p/T = \gamma$), there are difficulties in the analysis of the normal phase properties of this system. They arise from the closeness of its Curie temperature ($T_{\mathrm{Curie}} \sim 2.5\,\mathrm{K}$) to the superconducting transition ($T_{\mathrm{sc}} \sim 0.5\,\mathrm{K}$). For example, it is hard to determine precisely the $A$ coefficient of resistivity, because the temperature range for the $T$-square fit is not large enough. As regards the specific heat measurements, the $C_p/T$ values are significantly enhanced around the ferromagnetic transition. To obtain the Sommerfeld coefficient $\gamma$ correctly, we measured for each field the temperature dependence of $C_p/T$ from 1.6 K, down to the onset of the superconducting transition, to stay far from the $T_{\mathrm{Curie}}$ anomalie. Some of the $C_p/T$ curves at different fields are presented in Fig. 7a.

For magnetic fields close to 0 T, we fit the whole curve with an empirical law $C_p/T = \gamma + \delta.\exp(-T_0/T)$, in order to take into account the increase of $C_p/T$ above 1 K because of the proximity to $T_{\mathrm{Curie}}$. For magnetic fields above 0.1 T, where the anomalie at $T_{\mathrm{Curie}}$ almost disappears, the exponential term of the fit becomes negligible. $\gamma$ is then taken as the mean value of $C_p/T$ in the whole temperature range. The corresponding fits are presented with solid lines for each field in Fig. 7a.

Figure 7b shows the extracted field dependence of the Sommerfeld coefficient $\gamma(H)$ normalized to its zero field value $\gamma(0) = 0.056\,\mathrm{J\,K^{-2}\,mol^{-1}}$, for **H** along **c**-axis. On the same figure, $\gamma(H)/\gamma(0)$ for URhGe (from ref. 44) is shown for comparison, also for **H//c**.

### Strong coupling calculation of $H_{\mathrm{c2}}$.

The strong-coupling calculations for the upper critical field are based on a theoretical model described in ref. 9, where the Eliashberg equations are solved for a system of electrons and phonons with an Einstein spectrum. The calculations were performed in the clean limit for superconductivity. The orbital limitation is controlled by an averaged Fermi velocity $\langle v_F \rangle_\perp$ perpendicular to the applied field. The superconducting critical temperature $T_{\mathrm{sc}}$ is monitored by the strong-coupling parameter $\lambda$, the screened Coulomb repulsion parametrized by $\mu^*$, and an average frequency $\Omega$ for the pairing mechanism (analogue to the Debye frequency). The computation of $\mu^*$ remains a difficult problem even for simple metallic elements, but its value ranges typically between 0.1 and 0.15. In our calculation, $\mu^*$ was fixed at 0.1, and was considered to be independent of the magnetic field. The value of $\lambda$ at zero field was estimated by comparing the experimental $H_{\mathrm{c2}}$ with specific heat data, as explained in the text. The parameter $\Omega = 23.7\,\mathrm{K}$ was then adjusted to give the right $T_{\mathrm{sc}}$ at zero field. The averaged Fermi velocity $v_F \sim 2,600\,\mathrm{m\,s^{-1}}$ was adjusted to match the slope at $T_{\mathrm{sc}}$ of the experimental $H_{\mathrm{c2}}$ curve, for field along the **a** axis: in UCoGe, along this hard magnetization axis, the field dependence of $\lambda$ should be very small (experimentally $T_{\mathrm{Curie}}$ remains almost unchanged for field **H//a** up to 5 T). When the strong-coupling pairing strength $\lambda$ is varied, subsequent renormalization of the Fermi velocity $\langle v_F \rangle_\perp$ was taken into account, through equation (4). $\langle v_F \rangle_{\mathrm{band}}$ was considered to be field independent.

### Determination of $\lambda(0)$ in URhGe.

As explained in the main paper, the effects of the suppression of $\lambda$ under field in URhGe for **H//c** are much weaker, so the same method as used for UCoGe is not applicable. However, contrary to UCoGe, the specific heat has been estimated from magnetization measurements using Maxwell relations[44] for the three directions. At the field $H_R$ parallel to **b**, where the superconducting temperature is maximum, the specific heat coefficient is increased by more than 30%. If we believe that this increase of $C_p/T$ is because of the reinforcement of the pairing mechanism, this puts severe constraints on the value of $\lambda(0)$, as $\lambda(H) = \frac{\gamma(H)}{\gamma(0)}(1 + \lambda(0)) - 1$. So we used the data along the **b**-axis in URhGe, to adjust the value of $\lambda(0)$. Figure 8 shows the calculations used to derive $\lambda(H)$ for the optimal value $\lambda(0) = 0.75$, and Fig. 9 shows the comparison with the specific heat data, as well as the sensitivity to the value of $\lambda(0)$. It shows that the maximum value of the specific heat coefficient gives a selective criterium to fix $\lambda(0)$. With $\lambda(0) = 0.75$, $\lambda(H)$ increases almost by a factor 2, up to 1.4 at $H_R$ (see inset of Fig. 9): this puts URhGe in the strong-coupling limit, and it is a mechanical consequence of the large increase of the specific heat coefficient for **H//b** when it is attributed to the field variation of the pairing mechanism. The values of $\lambda$ along the **b** and **c** direction have been derived, as for UCoGe, using a field independent average frequency for the pairing mechanism $\Omega = 5.3\,\mathrm{K}$, adjusted to match $T_{\mathrm{sc}} = 0.26\,\mathrm{K}$ at zero field, with a screened Coulomb repulsion parameter $\mu^* = 0.1$. The renormalized Fermi velocity for $\lambda = \lambda(0)$ is $\sim 3,100\,\mathrm{m/s}$ for **H//b**, and $\sim 3,700\,\mathrm{m/s}$ for **H//c**. However, the real values might be different, as a calculation for a $p$-wave state also includes an anisotropy coming from the gap anisotropy, which has not been taken into account in these calculations[35,46].

### Data availability.

The data that support the findings of this study are available from the corresponding author on reasonable request.

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

## Acknowledgements

We thank V. Mineev for his help and reactivity on the analysis of our results. We also thank M. Zhitomirskii, G. Knebel, A. Pourret, J. Flouquet, Y. Tokunaga for useful discussions, and A. Gourgout for experimental support. The work was supported by the ERC NewHeavyFermions, the programs KAKENHI (15H05882, 15H05884, 15K21732, 16H04006, 15H05745), and the ANR grant SINUS.

## Author contributions

D.A. prepared the crystals, L.H. and M.T. performed the first measurements of thermal and electrical transport, which were extended to high field by B.W., who also performed the specific heat measurements. C.P. did the magnetization and ac susceptibility measurements. Data analysis and interpretation were done by B.W., G.B. and J.-P.B. who supervised the work. The paper was written by B.W., and J.-P.B. with inputs from all authors.

## Additional information

**Competing financial interests:** The authors declare no competing financial interests.

**Publisher's note**: 

