## [Peer Review File · Nature Communications]

Reviewers' comments:

Reviewer #1 (Remarks to the Author):

Referee report on

Manuscript NCOMMS-16-20590-T

"Pairing mechanism in the ferromagnetic superconductor UCoGe"

by Beilun Wu et al.

The present manuscript deals with a central question in a field of high current interest in condensed-matter physics, i. e., the pairing mechanism in the ferromagnetic superconductor UCoGe. The authors present a very careful and convincing analysis of the highly unusual experimental properties of this material. The suggested procedure will stimulate the discussion and certainly open new avenues of research in the field.

For these reasons, I strongly recommend publication of the manuscript in its present form.

Reviewer #2 (Remarks to the Author):

This paper reports the field dependence of the pairing interaction in the ferromagnetic superconductor UCoGe, which is used for understanding quantitatively the unusual behaviors of the upper critical field of this system, and provides the evidence of a dominant role of ferromagnetic spin fluctuations in the pairing mechanism. The authors carried out the thermal transport measurements to determine the upper critical fields. Then, they presented a model of the field-dependent pairing interaction which leads to the good agreement between the calculated upper critical fields based on this model and the experimental data. The determination of the upper critical fields of UCoGe based on thermal transport measurements is new. However, the idea that the field-dependent pairing interaction mediated via ferromagnetic spin fluctuations can explain the unusual behaviors of the upper critical fields in UCoGe has already been presented in refs.18 and 28, and hence, is not new. In refs. 18 and 28, it was discussed and clarified that the strong suppression of the upper critical field for the field along the c-axis is due to the weakening of the pairing interaction caused by the applied magnetic field along the easy axis, which suppresses the longitudinal ferromagnetic spin fluctuation. The main difference between the model of the current paper and that in the previous papers seems to be just the precise functional form of the pairing interaction with respect to the magnetic field; i.e. the former is determined from the phenomenological model proposed in ref.29 and the fitting to the data of the specific heat measurements, and the latter is based on the NMR measurement results. Although it may be important to resolve the discrepancy of the precise field dependence, the overall behaviors, i.e. the suppression of the pairing interaction due to the magnetic field parallel to the c-axis, are similar. Thus, I feel that the novelty of the idea of this paper is rather limited.

Also, the analysis of the effective mass presented in Eq.(4) is a bit crude, because this formula can not

capture effects of critical spin fluctuations on mass enhancement correctly. Thus, it is not sure whether the coincidence with the experimental data of the specific heat shown in Fig.3 actually supports the validity of the model. Because of the above reasons, I cannot recommend the paper for publication in Nature Communications in the current form.

Reviewer #3 (Remarks to the Author):

The ferromagnetic superconductor UCoGe is well-known as a promising candidate of the triplet pairing formation of Cooper pairs. Recently, it was revealed by NMR experiments that the superconductivity is mediated by longitudinal ferromagnetic fluctuations. In this manuscript, the authors measured the thermal conductivity, specific heat, magnetization, and the ac magnetic susceptibility, which allows the authors to test the hypothesis of a field-dependent pairing mechanism. Comparing their experimental results with a theory, they succeeded in showing that some properties like the unusual temperature-dependence and anisotropy in the superconducting upper critical fields (along the magnetization easy-axis) can be understood within a theoretical framework describing superconductivity arising from ferromagnetic fluctuations. In particular, the excellent agreement as shown in Fig.3 is very impressive for me. These results strongly suggest the strong variation of the pairing interaction with the magnetic field, and demonstrate the relationship between the superconducting and normal-state properties. This paper is helpful for a deeper understanding of the correlation between superconductivity and magnetism, and has an immediate impact on the researchers in the strongly correlated electron physics as well as superconductivity. As a result, I recommend the publication of this manuscript in Nature Communications.

However, prior to publication, the authors should consider the following comments:

- (1) The specific heat was measured, but the data at around T_{SC} are not presented in the draft: I wonder if the T_{SC} values are consistent with those determined from the thermal conductivity measurements.
- (2) In the figure caption of Fig.4, we find the phrase "(see insert)". But, "insert" is missing in this figure.
- (3) There are careless mistypes in the manuscript.

REVIEWERS' COMMENTS:

Reviewer #3 (Remarks to the Author):

I feel that the points raised in my previous review have been satisfactorily addressed.

Dear Editor,

First, we would like to thank the referees for their careful reading of the manuscript, and we apologize for the late answer. We include below a response to their remarks or question, and detail the corrections done on the manuscript.

-Reviewer #1: we thank the referee for his strong support to the publication of our work.

-Reviewer #2: by contrast to the other two referees, reviewer#2 was not convinced by the “novelty” of our results, as regards the analysis, pointing out the suppression of the pairing mechanism by an applied field along the c-axis. We quote here his first criticism:

“However, the idea that the field-dependent pairing interaction mediated via ferromagnetic spin fluctuations can explain the unusual behaviours of the upper critical fields in UCoGe has already been presented in refs. 18 and 28, and hence, is not new. In refs. 18 and 28, it was discussed and clarified that the strong suppression of the upper critical field for the field along the c-axis is due to the weakening of the pairing interaction caused by the applied magnetic field along the easy axis, which suppresses the longitudinal ferromagnetic spin fluctuation.”

As we said in the paper, reference 18 has clearly established, with $1/T_1$ measurements, that spin fluctuations in UCoGe are Ising type, and strongly suppressed by the c-component of the applied magnetic field. As we also wrote, they proposed a first model for this system, to explore what this could imply for H_{c2} . But we disagree with the point of view that it “clarified” that the small value of H_{c2} along the c-axis is due to this suppression.

- First of all, the idea that the pairing interaction in ferromagnetic superconductors could be field dependent is indeed not new, and was proposed even before the NMR papers (references 18 and 28): for example, it was early on suggested that the re-entrant phase in URhGe was due to an enhancement of the ferromagnetic fluctuations at the proximity of a field induced quantum critical point (see Levy, F et al., “Acute enhancement of the upper critical field for superconductivity approaching a quantum critical point in URhGe”, *Nature Phys.* **2007**, 3, 460-463), and the idea that the pairing should be suppressed for H along the easy-axis was published explicitly by V. Mineev in 2011 (V. Mineev, “Magnetic field dependence of pairing interaction in ferromagnetic superconductors with triplet pairing”, *PRB* **2011**, 83, 064515).

- The whole question however, is to know if this concept does apply to UCoGe, and if it explains the peculiar behaviour of H_{c2} .

So the following comment of the referee miss what we believe is the real novelty of our paper:

« The main difference between the model of the current paper and that in the previous papers seems to be just the precise functional form of the pairing interaction with respect to the magnetic field; i.e. the former is determined from the phenomenological model proposed in ref.29 and the fitting to the data of the specific heat measurements, and the latter is based on the NMR measurement results. Although it may be important to resolve the discrepancy of the precise field dependence, the overall behaviors, i.e. the suppression of the pairing interaction due to the magnetic field parallel to the c-axis, are similar. Thus, I feel that the novelty of the idea of this paper is rather limited.»

From our point of view, the analysis we present in the paper is new in the following aspects:

- We have explained how the field suppression of λ can explain the positive curvature of H_{c2} , which is counter-intuitive. This of course emerges naturally from a complete microscopic calculation, and so was included in the calculations of references 18 and 28, but with no clue to understand in which case it could work, and how to check it.

- We have used a new analysis of H_{c2} , proposing to extract the required behaviour from

the data rather than finding the model that would fit the data: this allows comparison of the “bare data” in the superconducting phase, with normal state properties, through the field dependence of the strong-coupling parameter λ . And thanks to the simple explanation mentioned above for the relation between $\lambda(H)$ and H_{c2} curvature, it provides a first quantitative support for the explanation of the H_{c2} suppression from the field dependence of λ , independent of the explanation for the origin of this field dependence.

- We use an explicit expression (from references 14 and 29) for the field dependence of the pairing, derived from a well-justified theoretical background. All the parameters for the field variation of λ have a clear significance, and we could compare or deduce them from experimental measurements. This is particularly rare for strongly correlated electron systems, and it is pivotal for the identification of the ferromagnetic fluctuations as the pairing mechanism. By contrast, in references 18 and 28, the expression of the susceptibility is “inspired” by the field dependence of the fluctuations which seem to be suppressed at a rate going like \sqrt{B} . But there is no theoretical justification for this power law. A fortiori, there is also no way to check if the field dependence observed by NMR can explain the amplitude of the effect put in the susceptibility, so as to reproduce the order of magnitude of the H_{c2} anisotropy at low temperature.

- We could explained the difference between the two systems UCoGe and URhGe, thanks to the deep understanding of the origin of the field dependence of the pairing, and to the fact that we are quantitative. This was not possible for the NMR work, which did not say a word of the case of URhGe.

Let us note also that in the references 18 and 29, there is no comparison of H_{c2} data with the calculations. Quantitatively, even with uncontrolled parameters, it would not have worked, due to the \sqrt{B} dependence, which results in an infinite anisotropy between c and a or b-axis: the NMR work is clearly an important work, but it is different from what we present.

So we hope that the referee will be able to revise his judgements, and understand that our work does really bring a new (and strong!) support for the identification of the pairing mechanism in UCoGe.

The last point raised by the reviewer is not completely clear to us:

“Also, the analysis of the effective mass presented in Eq.(4) is a bit crude, because this formula can not capture effects of critical spin fluctuations on mass enhancement correctly. Thus, it is not sure whether the coincidence with the experimental data of the specific heat shown in Fig.3 actually supports the validity of the model.”

Equation (4) is clearly valid in the zero temperature limit, or at least much below T_{Curie} , when Fermi liquid behaviour has been recovered. So indeed, it does not capture the effects of critical spin fluctuations on the specific heat, but this does not matter:

- Superconductivity is controlled by “virtual spin fluctuations”, not by the critical fluctuations at $T_{\text{Curie}} \gg T_{\text{sc}}$: what matters is the susceptibility at T_{sc} (and below).

- Experimentally, at T_{sc} (0.5K), we are far from the critical region, and as explained in the supplementary (see also figure S3 of the supplementary), we took great care to extract the Sommerfeld coefficient away from the critical fluctuations (their influence is observed at zero field above 1K), to compare with equation (4).

Moreover, such a formula is derived from microscopic models for spin fluctuations in metals, and is routinely used as can be seen for example in the work of Monthoux and Lonzarich (Magnetically mediated superconductivity in quasi-two and three dimensions, PRB **2001**, 63, 054529)

-Reviewer #3: we thank the referee for his strong support to the publication of our work. He made 3 comments:

(1) The specific heat was measured, but the data at around T_{SC} are not presented in the draft: I wonder if the T_{SC} values are consistent with those determined from the thermal conductivity measurements.

The reason why the data for the superconducting transition as measured by specific heat (C_p) are not presented in the work is technical: it was too difficult to perform the measurements at low temperature under field on this particular sample. Indeed, the sample is bar shaped cut along the c-axis for transport measurements. This means that only a small area of the sample is in contact with the specific heat set-up, if we want to apply the field along the c-axis. We could measure the superconducting transition up to 1T above 0.5K, and check in zero field that the superconducting transition determined with an “equal entropy construction” does match the thermal conductivity (and susceptibility) determinations. However, at lower temperatures, the measurements under field are difficult due to the sample geometry, which leads to long diffusion time constants. We did these measurements on another sample with a “platelet” shape, and the H_{c2} determined by C_p is very similar to that presented on the sample of the paper. But the shape of this second sample would lead to very inhomogeneous demagnetization fields, and so to an un-precise comparison of theory to experiments. As it seems really important in UCoGe, to compare measurements on the same sample (both T_{sc} and T_{Curie} are sensitive to defect, exact stoichiometry, impurities...), we preferred not to show the data on the other sample, to avoid confusion.

(2) In the figure caption of Fig.4, we find the phrase “(see insert)”. But, “insert” is missing in this figure.

This is corrected, thank you !

(3) There are careless mistypes in the manuscript.

We have read again the manuscript and corrected as much as we could!

Dear Editor,

After re-submission, we only received the following comment from referee #3:

REVIEWERS' COMMENTS:

Reviewer #3 (Remarks to the Author):

I feel that the points raised in my previous review have been satisfactorily addressed.

So we do not have anything else to add.